PLOS · Biology

# Long-term musical training can protect against age-related upregulation of neural activity in speech-in-noise perception

Lei Zhang[1,2,3], Bernhard Ross[3], Yi Du[1,2,4]*, Claude Alain[3,5,6,7]*

1 State Key Laboratory of Cognitive Science and Mental Health, Institute of Psychology, Chinese Academy of Sciences, Beijing, China, 2 Department of Psychology, University of Chinese Academy of Sciences, Beijing, China, 3 Rotman Research Institute, Baycrest Academy for Research and Education, Toronto, Canada, 4 Chinese Institute for Brain Research, Beijing, China, 5 Department of Psychology, University of Toronto, Toronto, Canada, 6 Institute of Medical Science, University of Toronto, Ontario, Canada, 7 Music and Health Science Research Collaboratory, University of Toronto, Toronto, Ontario, Canada

* duyi@psych.ac.cn (YD); calain@research.baycrest.org (CA)

## Abstract

During cognitive tasks, older adults often show increased frontoparietal neural activity and functional connectivity. Cognitive reserve accrued from positive life choices like long-term musical training can provide additional neural resources to help cope with the effect of aging. However, the relationship between cognitive reserve and upregulated neural activity in older adults remains poorly understood. In this study, we measured brain activity using functional magnetic resonance imaging during a speech-in-noise task and assessed whether cognitive reserve accumulated from long-term musical training bolsters or holds back age-related increase in neural activity. Older musicians exhibited less upregulation of task-induced functional connectivity than older non-musicians in auditory dorsal regions, which predicted better behavioral performance in older musicians. Furthermore, older musicians demonstrated more youth-like spatial patterns of functional connectivity, as compared to older non-musicians. Our findings show that cognitive reserve accrued through long-term music training holds back age-related neural recruitment during speech-in-noise perception and enlighten the intricate interplay between cognitive reserve and age-related upregulated activity during cognitive tasks.

## Introduction

Normal aging is typically associated with decline in sensory and cognitive functions. These age-related changes in perception and cognition are often accompanied by increased neural activity and functional connectivity in widely distributed neural networks, with these patterns varying depending on the stimuli and tasks used to investigate cognitive aging. The recruitment of neural activity and strengthening of

**Data availability statement:** The behavioral and fMRI data that support the findings of this study are available in OSF: https://osf.io/89hbn/.

**Funding:** This work was supported by STI 2030—Major Project (2021ZD0201500, https://service.most.gov.cn/) to YD, the Natural Sciences and Engineering Research Council (RGPIN-2021-02721, https://www.nserc-crsng.gc.ca/index_eng.asp) to CA, and Canadian Institute for Health Research (PJT 183614, https://cihr-irsc.gc.ca/e/193.html) to CA. The funders had no role in study design, data collection and analysis, decision to publish, or preparation of the manuscript.

**Competing interests:** The authors have declared that no competing interests exist.

**Abbreviations:** BOLD, blood-oxygen-level-dependent; fMRI, functional magnetic resonance imaging; GLM, general linear model; gPPI, generalized psychophysiological interaction; ISPC, inter-subject spatial correlation; IFGop, opercular part of inferior frontal gyrus; PrCGinf; inferior part of precentral gyrus; PrCGsup, superior part of precentral gyrus; pSTG, posterior superior temporal gyri; ROI, region of interest; RSFC, resting-state functional connectivity; SIN, speech-in-noise; SM, speech motor areas; SMA, supplementary motor area; SMG, supramarginal gyrus; SNRs, signal-to-noise ratios; SPL, sound pressure level; TiFC, task-induced functional connectivity.

functional connectivity are thought to reflect compensatory strategy employed by older adults to maintain optimal cognitive performance [1,2].

Current theories posit that older adults may automatically or voluntarily (e.g., spending more effort) engage in strategies to mitigate age-related declines in perception and cognition [3–7]. According to the Posterior–Anterior Shift in Aging and Compensation-Related Utilization of Neural Circuits model [8,9], older adults would exhibit upregulated neural activity in frontoparietal brain regions more extensively than younger adults during cognitive tasks [10] (Fig 1A). However, the degree of increased activity in these frontoparietal regions varies significantly among older adults and may be influenced by cognitive reserve [1,2,11]. The Scaffolding Theory of Aging and Cognition model suggests that positive lifestyle choices, such as musical training, higher levels of education, and bilingualism, contribute to cognitive and brain reserve, which represents the accumulation of cognitive and neural resources before the onset of age-related brain changes [1,6]. Thus, we hypothesized that individuals with greater cognitive reserve would demonstrate enhanced neural resources than those without such reserve (Fig 1B). Despite this, how cumulative reserve influenced by positive lifestyle factors impact the recruitment of neural activity in older populations remains controversial. Some studies have found that developing expertise is associated with reduced regional brain activity, which is explained by increased neural efficiency [1,12,13]. In contrast, others report stronger regional brain activity, attributed to greater functional capacity [14–16]. Notably, no studies to date have examined these effects in older adults.

In this functional magnetic resonance imaging (fMRI) study, we focused on musical training, as it requires sensory-motor integration and serves as an ideal model for investigating experience-dependent brain plasticity [17–20]. Moreover, musicians are known to exhibit enhanced cognitive and brain reserve particularly in auditory perception [14,16,21–26]. Prior research revealed enhanced sensorimotor integration, better speech signal encoding and more robust functional connectivity in the auditory dorsal brain region during speech processing [27–31]. These findings suggest that musicians possess greater neural resources than non-musicians, contributing to their enhanced task performance (Fig 1B). In comparison, older adults tend to over-recruit dorsal stream brain regions to compensate for deficits in speech-in-noise (SIN) perception (Fig 1A) [32–34]. Musicians showed greater activity and functional connectivity in dorsal stream regions when processing both music and SIN [14,21,35], indicating that musical training fosters cognitive reserve in these brain areas thereby easing SIN perception. These findings point to neuroplasticity in the auditory dorsal stream as a key factor underlying the impact of long-term musical training and aging on SIN perception. Therefore, we focus on the neural response within the auditory dorsal stream, which includes auditory, inferior parietal, dorsal frontal motor, and speech motor areas, supporting sound-to-action mapping and sensorimotor integration during speech processing [36,37].

Cognitive Reserve theory suggests that experiences, especially in childhood and throughout life, contribute to building a "reverse" of cognitive abilities, which can help mitigate the impact of age-related brain decline [38]. We propose two hypotheses

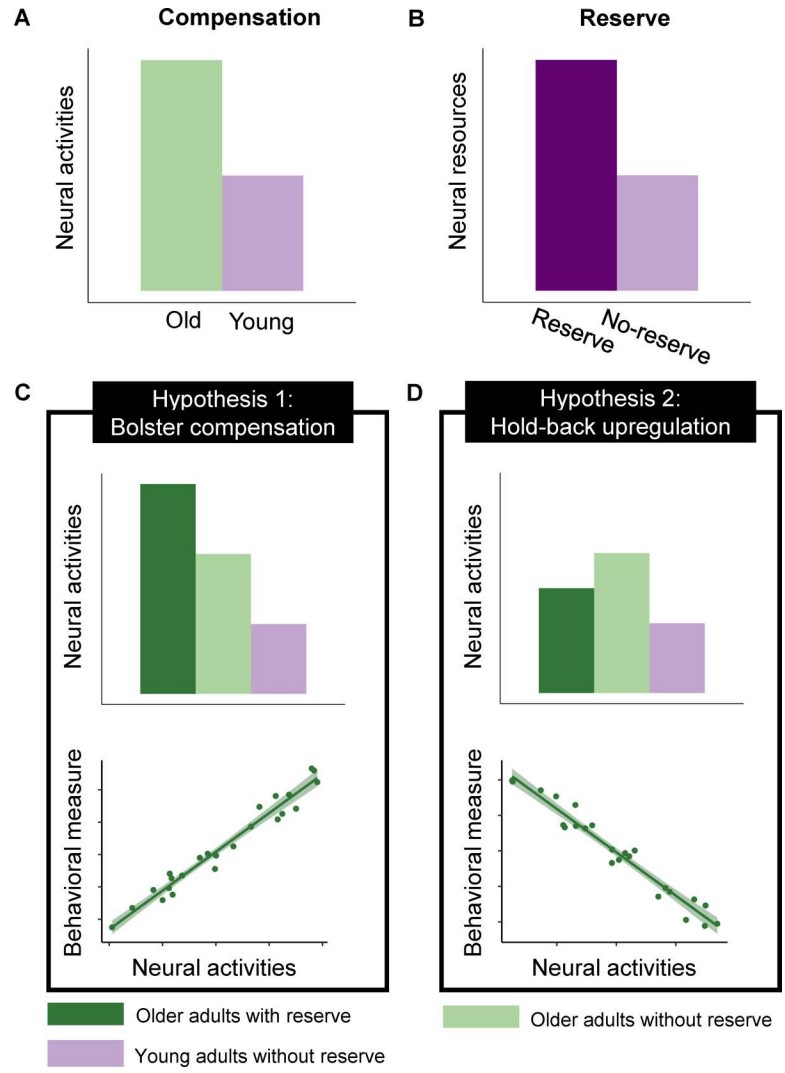

**Fig 1. Illustration of "Bolster compensation" and "Hold-back upregulation" hypotheses. A–B**, neural activity that supports age-related neural compensation and neural reserve. **C–D**, Hypothetical change in neural activities and brain-behavior correlation pattern due to the interplay between neural reserve and neural compensation.

regarding how accumulated reserve from long-term musical training interacts with upregulated neural activity when older adults perform SIN perception tasks. The "Bolster Compensation Hypothesis" (Fig 1C) suggests that brain reserve, as a cumulative enhancement of neural resources, strengthens compensatory upregulation of neural activity in older adults during cognitive tasks, with both mechanisms working to increase neural activity and support cognitive performance [1]. Conversely, the "Hold-Back Upregulation Hypothesis" (Fig 1D) posits that older adults with greater cognitive reserve display neural activity levels more comparable to those of younger adults, as cognitive reserve provides additional resources that mitigate an age-related decline in perception and cognition [6,7]. The more closely these activity levels resemble those of younger adults, the better the cognitive performance.

To test the proposed hypotheses, we invited 25 older musicians (OMs), 25 older non-musicians (ONMs), and 24 young non-musicians (YNMs) to identify syllables in noise (signal-to-noise ratio (SNR): −8, 0, +8 dB). We conducted

region of interest (ROI)-based generalized psychophysiological interaction (gPPI) analyses from bilateral auditory seeds (posterior superior temporal gyrus (pSTG)) to bilateral auditory dorsal stream regions (supramarginal gyrus (SMG), supplementary motor area (SMA), superior part of precentral gyrus (PrCGsup), speech motor areas (SM) including inferior part of precentral gyrus (PrCGinf) and opercular part of inferior frontal gyrus (IFGop)). We also examined the resting-state functional connectivity (RSFC) across the three groups to determine whether the hypotheses are supported solely by task-induced activity or also by intrinsic activity observed during the resting state. Additionally, we extracted blood-oxygen-level-dependent (BOLD) activation patterns in these ROIs, as most previous studies recognized neural compensation and neural reserve based on regional activations [1,2,32,33].

Observing upregulated neural activity in ONMs compared to OMs supports the "Hold-back upregulation hypothesis." In this case, we expect the level of neural activities in OMs to fall between those observed in ONMs and YNMs. We anticipate the difference between OMs and YNMs to be smaller than between ONMs and YNMs. We also expect a negative correlation between neural activities in OMs and their behavioral performance. However, an extra upregulation of the neural activities in OMs would support the "Bolster compensation hypothesis." This would be accompanied by a larger difference between OMs and YNMs than the difference between ONMs and YNMs and a positive correlation between neural activities in OMs and their behavioral performance.

## Results

### Musical training mitigated age-related declines in SIN perception

First, we examined whether long-term musical training protected against age-related declines in SIN perception among older adults by comparing the accuracy of OM, ONM and YNM groups in identifying syllables amidst background noise during fMRI acquisition. The two-way mixed-design analyses of variance (ANOVA) with group (OM/ONM/YNM) and SNR (8, 0, −8 dB) as between- and within-subject factor, respectively, revealed main effects of group and SNR, and an interaction between group and SNR (group: $F(2,71) = 28.65$, $P < 0.001$; SNR: $F(2,71) = 63.78$, $P < 0.001$; Interaction: $F(4,71) = 4.24$, $P = 0.004$). OMs showed better performance than ONMs under two higher SNR conditions (SNR 8: $t(71) = 3.89$, $P_{fdr} < 0.001$; SNR 0: $t(71) = 3.29$, $P_{fdr} = 0.002$; SNR −8: $t(71) = −0.26$, $P_{fdr} = 0.794$), albeit worse than YNMs (SNR 8: $t(71) = −3.15$, $P_{fdr} = 0.002$; SNR 0: $t(71) = −2.88$, $P_{fdr} = 0.005$; SNR −8: $t(71) = −5.59$, $P_{fdr} < 0.001$), and YNMs performed better than ONMs (SNR 8: $t(71) = 7.00$, $P_{fdr} < 0.001$; SNR 0: $t(71) = 6.13$, $P_{fdr} < 0.001$; SNR −8: $t(71) = 5.33$, $P_{fdr} < 0.001$) (Fig 2). Although the SNR effect in accuracy was significant, the differences among older adults, particularly non-musicians, across the three SNR levels were minimal (ONMs: 35%−43%; OMs: 34%−52%) and our hypotheses focuses on group differences. Therefore, we focus solely on the main effect of group in the following analyses. While we observed differences in years of education between older musicians and non-musicians, supplementary analyses of covariance to assess the effect of education confirmed that years of education did not significantly influence our behavioral or neural measures, ruling it out as a confounding factor (for details, see S1 Text).

### ONMs showed upregulated task-induced functional connectivity (TiFC) during SIN perception

We performed gPPI analysis to investigate how aging and musical training modulated the TiFC within the bilateral auditory dorsal streams, which are crucial in speech perception [36,37,39]. The gPPI analysis quantifies whether the task demands modulate functional connectivity between two brain regions [40]. According to the dual model of auditory processing [36,37,39] and previous results of speech perception studies [14,31,36,41], we used bilateral posterior STG as seeds in the auditory dorsal streams. Target ROIs included bilateral SMG, SMA, PrCGsup, and speech motor areas composed of PrCGinf and IFGop (Fig 3A, 3B). Two-way mixed-design ANOVAs (group × SNR) were performed to investigate the group effect on TiFC in the bilateral auditory dorsal streams. Main effect of SNR was not significant (all $P_{fdr} > 0.708$), while the interaction effect was significant in left hemisphere (all $P_{fdr} < 0.05$). Since our hypotheses focuses on group differences,

PLOS Biology

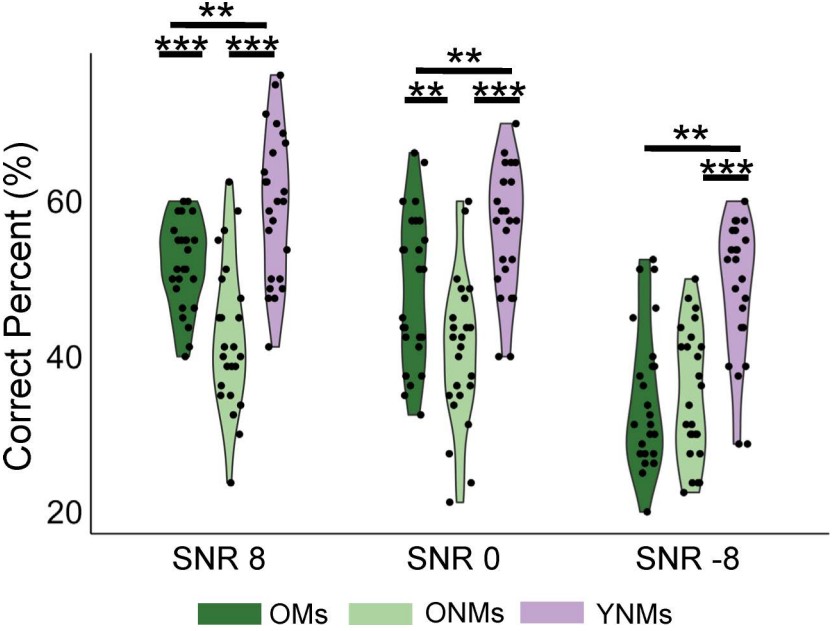

**Fig 2. Behavioral performance of the three groups under three SNR conditions.** Violin plots and individual data points of behavioral performance. OMs performed worse than YNMs under all SNRs, but better than ONMs under SNR 8 and SNR 0. Data are available on OSF (https://osf.io/89hbn/). ***$P_{fdr}$ < 0.001, **$P_{fdr}$ < 0.01. OMs, older musicians; ONMs, older non-musicians; YNMs, young non-musicians.

the interaction effect is not further analyzed. Outliers, defined as data points beyond two standard deviations, were replaced with the groups' median.

We found a main effect of group on TiFC within the bilateral auditory dorsal streams (LpSTG-LSMA, LSMG, LPrCGsup, LSM; RpSTG-RSMA, RSMG, RPrCGsup, RSM, for voxel-wise functional connectivity map of the three groups, see S2 Fig). For connectivity that showed a significant group effect after FDR correction, pairwise comparisons revealed stronger TiFC in ONMs than young adults (LpSTG-LSMA, LSMG, LPrCGsup, LSM; RpSTG-RSMA, RSMG, RPrCGsup, RSM, Fig 3C, S1 Table, for voxel-wise analysis, see S3A, S3C Fig).

Additionally, we employed robust linear mixed-effects analysis to investigate the effect of group on the left and right hemisphere TiFC after controlling for participants and brain regions. We found stronger overall TiFC in ONMs than YNMs (left: $\beta_{group}$ = 0.56, $P$ < 0.001; right: $\beta_{group}$ = 0.45, $P$ < 0.001).

## Connectivity strength in OM was more akin to YNM than ONM

Both OMs and ONMs showed stronger connectivity than YNMs in the left hemisphere (LpSTG-LSMA, LSMG, LPrCG-sup, LSM, Fig 3C, for statistics, see S1 Table, for voxel-wise analysis, see S3A, S3B, S3C Fig). However, OMs showed more youth-like TiFC than ONMs in certain ROIs (LpSTG-SMA, LSMG, LPrCGsup). Because standard parametric tests (e.g., ANOVA or linear mixed models) do not directly assess whether the magnitude of the difference between ONMs and YNMs exceeds that between OMs and YNMs, we employed a nonparametric bootstrap approach with 10,000 iterations (for details, see "Methods"). We found that the difference between OMs and YNMs across the left ROIs was significantly smaller than the difference between ONMs and YNMs (95% Bootstrap CI: 0.59, 1.21).

The ONMs showed stronger connectivity in the right hemisphere than OMs (RpSTG-RSMA, RSMG, RPrCGsup, for voxel-wise analysis, see S3D Fig) and YNMs (RpSTG-LSMA, RPrCGsup, RSMG, RSM). Still, differences between OMs and YNMs were not significant except for the connectivity between RpSTG and RSMA, as well as RpSTG and RSM.

PLOS Biology

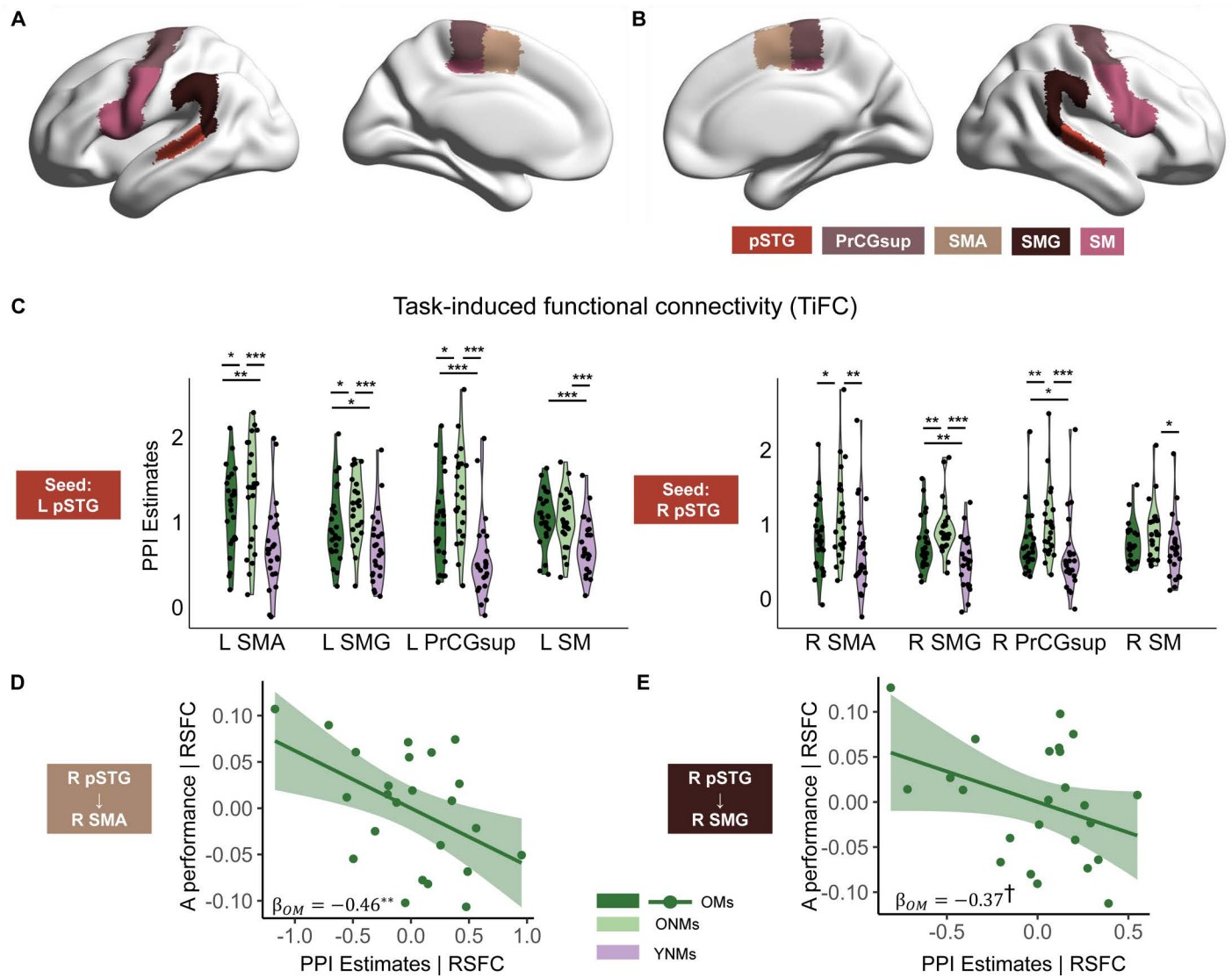

**Fig 3. TiFC of the three groups, and brain-behavior correlations. A**, left and **B**, right ROIs used in PPI analysis. Bilateral posterior superior temporal gyri (pSTG) were the seed ROI. Bilateral auditory dorsal stream regions, including supramarginal gyrus (SMG), supplementary motor area (SMA), superior part of precentral gyrus (PrCGsup), and speech motor areas (SM) were the target regions. **C**, ONMs showed upregulated task-induced functional connectivity (TiFC) in bilateral dorsal streams compared to YNMs, while OMs exhibited TiFC resembling YNMs. Violin and point plots show the individual TiFC in specific target regions. **D–E**, TiFC strength of OMs was negatively correlated to the behavioral performance after controlling for resting-state functional connectivity (RSFC). ROIs were mapped onto the brain surface using the BrainNet Viewer toolbox [42]. Data are available on OSF (https://osf.io/89hbn/). ***$P_{fdr}$ < 0.001, **$P_{fdr}$ < 0.01, *$P_{fdr}$ < 0.05, †$P_{fdr}$ < 0.1; OMs, older musicians; ONMs, older non-musicians; YNMs, young non-musicians; L, left; R, right.

We also found that the difference between OMs and YNMs across the right ROIs was significantly smaller than between ONMs and YNMs (95% Bootstrap CI: 0.35, 0.95). Additionally, with a linear mixed model, regardless of brain regions, we found stronger TiFC in ONMs than OMs ($\beta_{group}$ = 0.21 $P$ = 0.023) in the right hemisphere. OMs also showed stronger TiFC than YNMs ($\beta_{group}$ = 0.23 $P$ = 0.021).

Moreover, we observed a negative correlation between TiFC strength and behavioral performance in OMs in the right SMA ($\beta_{OMs}$ = −0.46, $P$ = 0.008, Fig 3D) and a marginally negative correlation in the right SMG ($\beta_{OMs}$ = −0.37, $P$ = 0.078,

), when resting-state functional connectivity (RSFC) was controlled as a covariate. In the right PrCGsup, the TiFC-behavior correlation was also negative but not significant ($\beta_{OMs}$ = −0.32, $P$ = 0.163). No other correlations were found. The connectivity strengths of OMs who showed better behavioral performance were more akin to those of young adults, especially in the right hemisphere. Thus, long-term musical training provides the cognitive reserve to mitigate age-related declines in SIN perception. The TiFC strength supports the "Hold-back upregulation" hypothesis.

## OMs showed higher spatial alignment of TiFC to YNMs than ONMs

We performed inter-subject spatial correlation (ISPC) to further explore whether OMs also showed more youth-like TiFC in fine spatial patterns in the regions exhibiting significant or marginally significantly lower TiFC strength than ONMs. ISPC measures the spatial neural alignment of multivoxel TiFC spatial patterns of the older brain to that observed in younger brains (Fig 4A). We found that OMs showed higher ISPC to YNMs than ONMs in left PrCGsup ($t(46)$ = 2.74, $P_{fdr}$ = 0.044, Fig 4B). That is, OMs showed both more youth-like TiFC strength and spatial TiFC pattern than ONMs. More importantly,

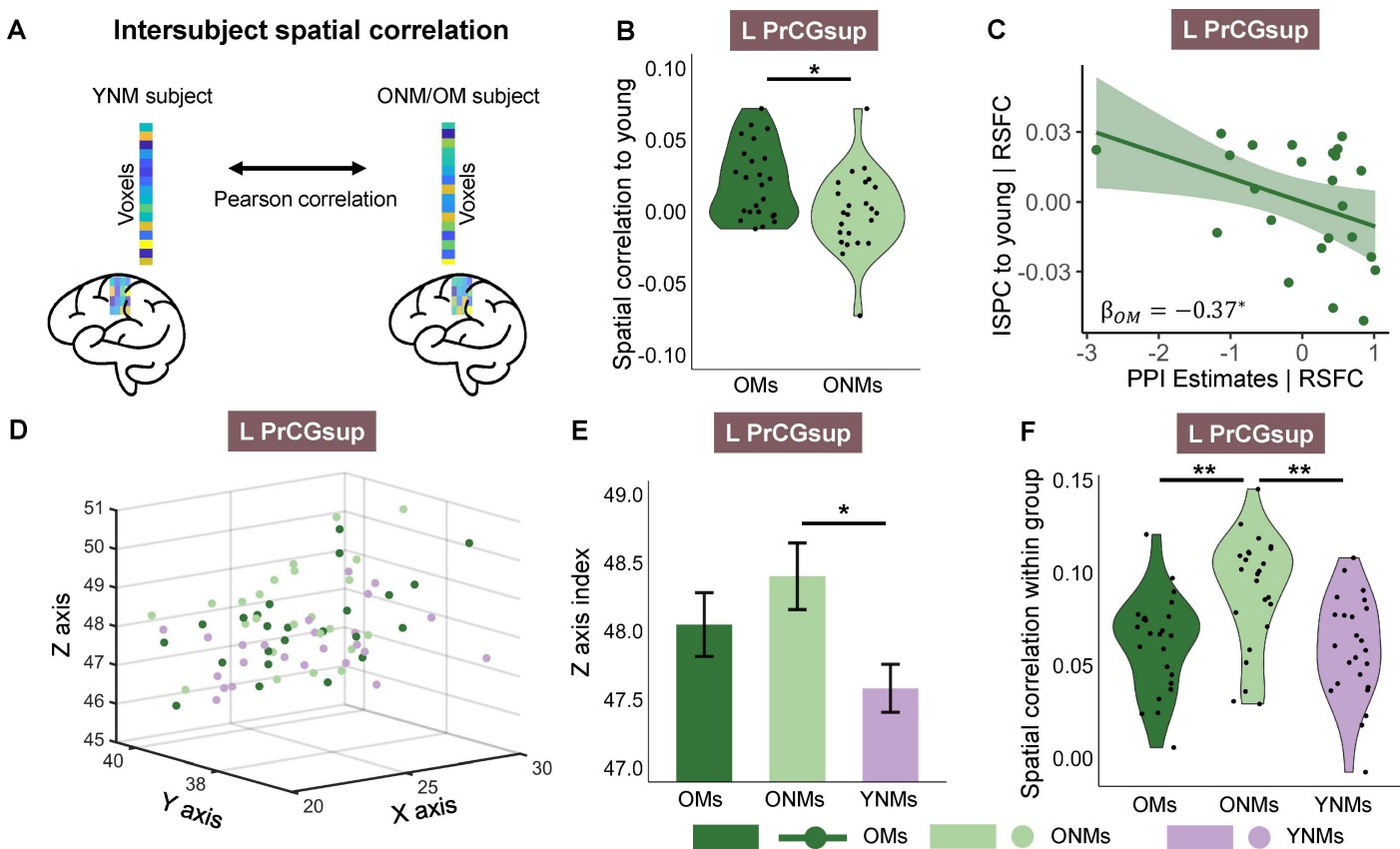

**Fig 4. Spatial alignment of TiFC to YNMs and within group. A**, Illustration of the calculation of intersubject spatial correlation; see "Methods" for details. **B–C**, OMs showed greater spatial alignment to YNMs than ONMs in left PrCGsup (B), and greater spatial alignment to YNMs predicted lower TiFC strength in OMs (C). **D–E**, Mean 3D coordinates of the voxels that showed top 10% TiFC were plotted in D. Voxels of ONMs exhibited significantly different positions in Z axis compared to YNMs, while no difference was found between OMs and YNMs (E). **F**, ONMs showed greater spatial correlation within the group than OMs and YNMs. Bar plots show the group mean of specific Z-axis index in L PrCGsup. Error bars indicate the SEM. Data are available on OSF (https://osf.io/89hbn/). Brain icon in panel **A** was sourced from: https://upload.wikimedia.org/wikipedia/commons/0/08/Wikimedia_Brand_Guidelines_Update_2022_-Brain.svg. *$P_{fdr}$ < 0.05, **$P_{fdr}$ < 0.01; OMs, older musicians; ONMs, older non-musicians; YNMs, young non-musicians; L PrCGsup, left superior part of precentral gyrus.

higher ISPC of OMs to YNMs negatively correlated with the TiFC strength after controlling for RSFC ($\beta_{OMs}$ = −0.37, $P$ = 0.029, Fig 4C). This suggests that the more youth-like spatial pattern is related to greater "Hold-back upregulation."

Next, we extended our analysis to quantify spatial shifts in connectivity peaks within the left PrCGsup. While ISPC measures overall spatial neural alignment between older and young adults, it does not capture specific reorganization patterns. By extracting the mean coordinates of the top 10% of voxels with the highest TiFC, we precisely mapped connectivity spatial shifts in older adults relative to young non-musicians (Fig 4D). We found a significant difference between ONMs and YNMs in the Z-axis direction ($t$(46) = 2.73, $P_{fdr}$ = 0.027, Fig 4E). Since the Z-axis increases from inferior to superior, the top voxels positions in ONMs were superior to those in YNMs. As expected, we found no difference between OMs and YNMs ($t$(46) = 1.60, $P_{fdr}$ = 0.175, Fig 4E).

We also calculated ISPC within groups. A higher within-group ISPC denotes that the TiFC spatial patterns are more similar within this group. We found that ONMs exhibited a higher ISPC than YNMs and OMs in left PrCGsup (YNM: $t$(46) = 3.71, $P_{fdr}$ = 0.002; OM: $t$(46) = 3.46, $P_{fdr}$ = 0.003, Fig 4F). However, no difference was found between OMs and YNMs ($t$(46) = 0.37, $P$ = 0.715, Fig 4F). In summary, ONMs' TiFC spatial pattern consistently deviated more from YNMs than OMs.

## Older adults showed stronger intrinsic RSFC in the auditory dorsal stream than young adults

We further examined whether the two hypotheses were supported by intrinsic functional connectivity itself by extracting RSFC in auditory dorsal stream regions seeded from bilateral pSTG. RSFC was measured by calculating Pearson correlation coefficients between voxel-wise time series in the same seed and target regions as used in the gPPI analysis. As illustrated in Fig 5, OMs demonstrated stronger RSFC than YNMs (LpSTG-LSMA, LSMG, LprCGsup; RpSTG-RSMA, RSMG, RPrCGsup, RSM, for statistics, see S2 Table), with no significant difference observed between OMs and ONMs. ONMs also showed stronger RSFC than YNMs (LpSTG-LPrCGsup; RpSTG-RPrCGsup, for statistics, see S2 Table).

However, we did not find any significant RSFC-behavior correlation in both OMs and ONMs groups. Therefore, age-related changes in intrinsic functional connectivity within the dorsal stream, as measured with RSFC, has little impact on task performance.

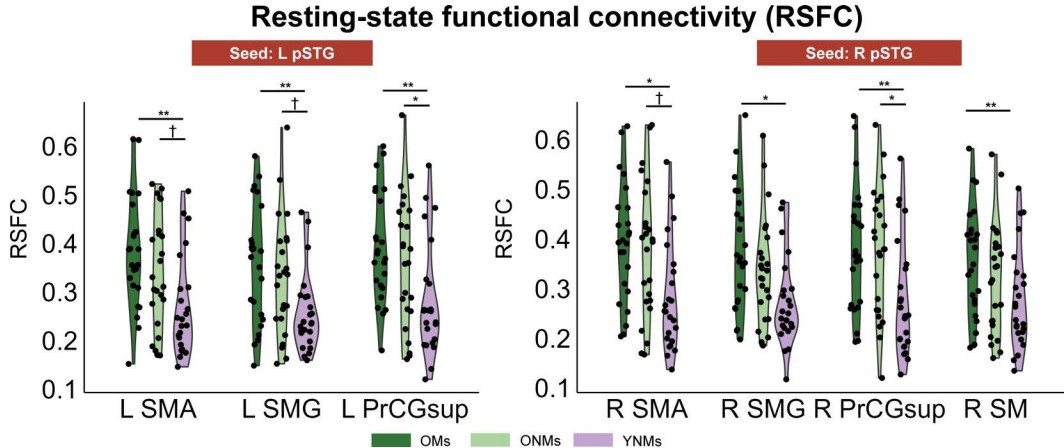

**Fig 5. Bilateral RSFC of the three groups.** ONMs and OMs showed upregulated RSFC in bilateral dorsal streams compared to YNMs. Violin and point plots show the individual RSFC in specific target regions. Data are available on OSF (https://osf.io/89hbn/). **$P_{fdr}$ < 0.01, *$P_{fdr}$ < 0.05, †$P_{fdr}$ < 0.1, OMs, older musicians; ONMs, older non-musicians; YNMs, young non-musicians; pSTG, posterior part of superior temporal gyrus; SMG, supramarginal gyrus; SMA, supplementary areas; SM, speech motor areas; PrCGsup, superior part of precentral gyrus; L, left; R, right.

**No group difference in BOLD activation was found**

We extracted BOLD activation in auditory dorsal stream regions to assess the two hypotheses regarding regional activation levels. However, we did not observe any BOLD activation difference between the three groups (for statistics, see S3 Table) or significant interaction effect between SNR and groups (all $P_{fdr}$ > 0.981). Voxel-wise analysis revealed significant activation in auditory dorsal stream regions from all three groups (S2 Fig), while the group difference in regional activation was not significant.

## Discussion

This study reveals how cognitive reserve accumulated through long-term musical training interacts with age-related neural compensation during SIN perception. We found reduced age-related declines in SIN performance among OMs. As expected, ONMs showed the typical age-related compensatory upregulation of TiFC in bilateral auditory dorsal streams. Notably, OMs exhibited a connectivity pattern in bilateral auditory dorsal streams that resembled YNMs, with connectivity strength in the right dorsal stream correlating with SIN perception. These findings support our "Hold-back upregulation" hypothesis, where cognitive reserve from musical training promotes a more youthful functional connectivity pattern, leading to superior behavioral outcomes. By further investigating fine spatial alignment of TiFC, we found that OMs exhibited more youth-like spatial pattern of TiFC, particularly between LpSTG and LPrCGsup, whereas ONMs showed a more consistently deviating TiFC spatial pattern alignment to YNMs compared to OMs. Lastly, we found increased intrinsic RSFC in older adults but no significant brain-behavior correlation with RSFC. Overall, our findings shed light on the intricate interplay between age-related upregulated neural activity and cognitive reserve. Reserve accrued from long-term musical training holds back the age-related upregulation of neural activities both in terms of strength level and fine spatial pattern such that OMs present youth-like listening skills.

We observed an age-related decline in SIN perception, consistent with past research [21,43]. Findings from earlier studies suggested that increased frontal BOLD activity in older normal-hearing individuals may serve as a compensatory mechanism during SIN tasks [32,33]. However, other studies have reported conflicting findings [34,44–46]. In our study, ONMs did not show increased regional BOLD activation during SIN perception. Instead, they exhibited heightened TiFC strength within bilateral auditory dorsal streams. The interpretation of such upregulated neural activity depends on brain-behavior correlations: a positive correlation suggests neural compensation, while a negative correlation indicates neural dysfunction [1,2]. However, we did not find a consistent positive or negative relationship between higher connectivity strength and better behavioral performance in ONMs. Therefore, it remains to be determined whether the upregulation of functional connectivity within auditory dorsal streams in ONMs indicates a compensatory strategy.

By investigating the neural responses of OMs, we showed how accumulated cognitive reserve from long-term musical training interacts with age-related neural upregulation. Previous studies in young adults have found that musicians show greater TiFC from auditory to motor areas during SIN perception, supporting their superior SIN perception performance [14]. Additionally, musicians displayed enhanced sensorimotor integration due to intensive sensory-motor interaction during musical training [16,17,19,47]. In the both hemispheres, we observed that OMs showed a connectivity profile more akin to YNMs than ONMs, based on the mean TiFC strength in most regions and the group difference across regions. In addition, lower connectivity strength in the right dorsal stream predicts better behavioral performance in OMs. These results support the "Hold-back upregulation" hypothesis, suggesting that the cognitive reserve provided by long-term musical training preserves a youth-like brain connectivity strength, which contribute to better behavioral performance. This is consistent with our previous findings using the same dataset, which show that OMs maintain better SIN perception performance by preserving youth-like representations in sensorimotor regions [21]. Previous studies have found that young musicians exhibit both functional and structural advantages of the right dorsal stream compared to YNMs, which may explain the potentially greater "Hold-back upregulation" in the right hemisphere in OMs [14,48–50].

Cognitive Reserve theory posits that "reserve" accrued through experience and training can help cope with or compensate for life-course related brain changes, leading to better-than-expected cognitive performance [1,2,38]. Our results are consistent with this framework, as cognitive reserve derived from musical training helps older musicians maintain superior SIN perception performance relative to older non-musicians. Furthermore, our study expands cognitive reserve theory by providing evidence of a potential neural mechanism underlying its protective effects. Specifically, cognitive reserve derived from long-term musical training appears to help preserve a youth-like neural activity pattern in key auditory and sensorimotor regions, which is associated with better-than-expected SIN perception performance in older adults. This finding suggests that, beyond merely compensating for age-related declines, cognitive reserve may work by maintaining the integrity and functional architecture of neural networks, thereby mitigating the adverse effects of aging on cognitive performance. Future studies should further test this "Hold-back upregulation" hypothesis using different cognitive tasks, such as memory and attention tasks, and investigate other sources of reserve, such as physical exercise and bilingualism. Additionally, examining how different types of learning transfer may influence such interactions will be important for generalizing the findings.

Moreover, this study significantly extends prior research [21] by showing both averaged TiFC strength and fine spatial pattern of TiFC during SIN processing, collectively suggesting that cognitive reserve from long-term musical training mitigates age-related declines in SIN perception by preserving youth-like TiFC patterns within task-related neural networks. Previous studies on age-related changes in brain function have compared univariate activities and multivoxel representations of presented stimuli [1,2,32]. By correlating the spatial patterns between older and young adults, referred to as intersubject spatial correlation, we directly examine the similarity of fine activation patterns between age groups. Both prior and current studies have found that youth-like neural activity patterns are associated with preserved behavioral performance and neural activity levels in older adults [21]. We believe that quantifying the similarity of fine activity patterns between young and older brains provides a valid and intuitive index of functional aging. Furthermore, by quantifying the precise spatial shifts in connectivity peaks using the coordinate-based analysis, we extended our analysis beyond ISPC's alignment index. While the ISPC measure provided an overall index of spatial alignment, it could not reveal specific reorganization details. Consistent with previous study using a memory task [51], our findings suggest systematic age-related reorganization of functional connectivity, possibly due to alterations in white matter integrity between auditory and motor regions. The absence of this shift in OMs implies that long-term musical training may help maintain a youth-like spatial configuration of neural connectivity. This preserved spatial pattern, together with more youth-like overall connectivity strength, supports the "Hold-back upregulation" hypothesis, wherein cognitive reserve modulates both the magnitude and fine-scale topography of task-induced functional connectivity during SIN perception. However, additional structural evidence is needed to fully understand the neural substrate underlying these spatial shifts.

According to the dual-stream model of speech perception, the auditory dorsal stream supports sound-to-action mapping during speech processing [36,37]. Previous studies have shown that task-related neural activities increase in auditory dorsal stream regions, including the frontal motor areas (e.g., premotor cortex and IFG), to compensate for deteriorating auditory signals via sensorimotor integration [31,32,52]. Degraded auditory function due to aging can also be regarded as an adverse condition [53]. Learning to play a musical instrument is a long-term process that requires intensive sensorimotor integration among auditory, visual, and motor systems. Musicians have been shown to have an advantage in auditory-motor integration during SIN perception [14,21]. Therefore, the auditory dorsal stream is central to age-related upregulation and the neural reserve from musical training, making it ideal for investigating how these two mechanisms interact.

RSFC is an excellent approach for examining intrinsic connectivity of participants independent of specific tasks. Previous studies have found that musicians showed more robust RSFC than non-musicians in speech-related brain networks [54–56]. We also examined the intrinsic functional connectivity in the three groups and included RSFC as a covariate in the brain-behavior correlation analysis. We found that both ONMs and OMs showed stronger RSFC compared to YNMs. Previous research has found decreased within-network connectivity but increased between-network connectivity in older adults compared to young adults (for reviews, see [57]). This aligns with our findings that older adults' connectivity

between auditory and frontal motor areas increased. However, whether RSFC is necessarily correlated with behavioral performance remains unclear [55,58]. In our study, we did not find any significant RSFC-behavior correlation in either older group. Hence, it cannot be concluded that the stronger intrinsic functional connectivity observed in older adults serves as a compensatory mechanism. Further studies are needed to explore the relationship between RSFC and SIN perception in older adults.

Our study has some limitations. First, we cannot establish cause-and-effect relationships between cognitive reserve from musical expertise and age-related changes in SIN perception using a cross-sectional design. Second, the investigation focused solely on the cognitive reserve derived from musical training and its effect on one specific task, SIN perception. It is unclear whether our findings generalize to other sources of cognitive reserve and other cognitive functions. Moreover, our sample size was relatively modest, limiting statistical power. All OMs were pooled regardless of training type (e.g., piano, voice, violin). Different training modalities engage distinct sensorimotor networks. Singing primarily engages articulatory control, while piano performance recruits fine finger motor circuits. Future studies comparing older adults with type-specific expertise could clarify how different forms of musical training contribute to cognitive reserve in SIN perception. Future studies should aim to use larger, more diverse samples, longitudinal designs, and alternative task paradigms to further elucidate the interplay between cognitive reserve and age-related neural changes. Future studies should explore how varying task difficulty (e.g., different SNRs) interacts with cognitive reserve to shape age-related changes in neural activity.

In conclusion, this fMRI study offers new insights into the intricate interplay between neural reserve and age-related upregulated activity in SIN perception among older adults. Specifically, it examines the newly proposed "Hold-back upregulation" and "Bolster compensation" hypotheses within this context. Our findings suggest that long-term musical training mitigates age-related decline in SIN perception by enhancing cognitive reserve, which interacts with upregulation of tasked-induced functional connectivity in ways consistent with the "Hold-back upregulation" hypothesis. Our results underscore the importance of considering both cognitive reserve and age-related upregulated neural activity in understanding cognitive performance, like SIN perception, in older adults. These findings may inform interventions aimed at preserving cognitive function and improve communication outcomes in aging populations.

## Methods

### Ethics statement

All participants provided written consent before taking part in the study, which was approved by the ethics committee of the Institute of Psychology, Chinese Academy of Science (Approval No. H16040). The study has been conducted according to the principles expressed in the Declaration of Helsinki.

### Participants

Twenty-five OMs, 25 ONMs, and 24 YNMs participated in this study. One OM was excluded from the fMRI analysis because of excessive head movements, and another ONM because of left-handedness. All remaining participants were healthy, right-handed, native Mandarin speakers with no history of neurological disorder and normal hearing (average pure tone threshold <20 dB hearing level between 250 and 4,000 Hz) in both ears. All older adults passed the Beijing version of the Montreal Cognitive Assessment (MoCA, ≥26 scores) [59]. All OMs started musical training before the age of 23 years (10.90±4.56 years old) and had at least 32 years of musical training (50.88±8.75 years). Also, all OMs practiced regularly in the last three years (12.70±8.99 hrs per week). All non-musicians reported less than two years of musical expertise experience. The demographic information of the three groups is summarized in Table 1. OMs and ONMs were not different in age, pure tone average, and MoCA score. However, self-reported education years differed between groups (OMs>ONMs, $t=4.15$, $P<0.001$). We conducted supplementary analyses to control for the effect of years of education on our results (see S1 Text of the Supplementary Materials).

**Table 1. The group mean (standard deviation) values and statistics of age, education, pure tone average (PTA) at 250–4,000 Hz, MOCA score, age of training onset, and years of music training in each group.**

| Group | Age | Edu | PTA | MOCA | Age of onset | Years of training |
|---|---|---|---|---|---|---|
| Older musicians | 65.12 (4.06) | 13.02 (2.96) | 12.38 (5.39) | 27.92 (1.19) | 10.90 (4.56) | 50.88 (8.75) |
| Older non-musicains | 66.64 (3.40) | 9.50 (3.02) | 12.58 (3.58) | 27.52 (1.39) | NA | NA |
| t (p) | −1.43 (0.158) | 4.15 (<0.001) | −0.15 (0.881) | 1.09 (0.28) | | |
| Young non-musicains | 23.13 (2.38) | 16.50 (1.67) | 0.71 (3.32) | NA | NA | NA |

## Stimuli and procedure

The stimuli comprised four naturally pronounced consonant-vowel syllables (/ba/,/da/,/pa/,/ta/) uttered by a 23 years old Chinese female. The utterances were recorded in a soundproof room. The syllable stimuli were about 400 ms in duration, low-pass filtered (4 kHz), and matched for average root-mean-square sound pressure level (for more details, see [21,31]). The masker was a speech spectrum-shaped noise (4-kHz low-pass, 10-ms rise-decay envelope) representative of 113 sentences by 50 young Chinese female speakers (age range 20–26 years). The stimuli were delivered at a level of 90 dB sound pressure level (SPL). The SPL of the masking sounds was modified to create three signal-to-noise ratios (SNRs) of −8, 0, and 8 dB. The stimuli were transmitted using MRI-compatible insert earphones (S14, Sensimetrics Corporation), which attenuated the scanner noise by up to 40 dB.

In the fMRI scanner, participants listened to the speech signals and identified the syllables by pressing the corresponding button using their right-hand fingers (index to little fingers in response to ba, da, pa, and ta in half of the participants or pa, ta, ba and da in the other half of the participants sequentially). Each participant completed four blocks. Each block contained 60 stimuli trials (3 SNRs × 4 syllables × 5 repetitions). As shown in S1A Fig, each trial presented a 400 ms syllable (see waveform examples in S1B Fig) followed by a jittered intertrial interval of 4–6 s (0.5 s steps). Four syllables under 3 SNRs were pseudo-randomly ordered within blocks.

## Behavior data analysis

Mixed-design ANOVAs were performed to investigate the main effect of group on behavioral performance collected during fMRI scanning. Greenhouse–Geisser correction was applied if the sphericity assumption was violated. FDR correction accounted for multiple comparisons in post hoc analysis. Statistical analyses were performed with the package bruceR [60] in R.

## Imaging data acquisition and preprocessing

Functional imaging data was collected using a 3T MRI system (Siemens Magnetom Trio), and T1 weighted images were acquired using the MPRAGE sequence (TR = 2,200 ms, TE = 3.49 ms, FOV = 256 mm, voxel size = 1 × 1 × 1 mm). Blood oxygen level-dependent images were acquired using the multiband-accelerated EPI sequence (acceleration factor = 4, TR = 640 ms, TE = 30 ms, slices = 40, FOV = 192, voxel sizes = 3 × 3 × 3 mm).

Task-state functional imaging data were preprocessed using AFNI software [61]. The first eight volumes were removed. The following preprocessing steps included slice timing, motion correction, aligning functional images with anatomy, spatial normalization (MNI152 space), spatial smoothing with 6-mm FWHM isotropic Gaussian kernel, and scaling each voxel time series to have a mean of 100.

## Univariate general linear model (GLM) analysis

Single-participant multiple-regression modeling was performed using the AFNI program 3dDeconvolve. Four syllables under three SNRs and six regressors corresponding to motion parameters were entered into the analysis. TRs would be

censored if the motion derivatives exceeded 0.3. For each SNR, the four syllables were grouped and contrasted against the baseline, and BOLD activation was averaged across SNRs.

## Task-induced functional connectivity

The gPPI analysis was used to examine the TiFC between auditory seeds and other regions in the auditory dorsal stream [40]. Auditory seeds were defined as bilateral posterior division of STG labels from the Harvard–Oxford atlas. Target regions were auditory dorsal stream regions, including bilateral SMG, SMA, PrCGsup, and SM area, which comprises preCGinf and IFGop [31,36,41]. Note that since inter-hemispheric connectivity was not of interest in this study, only intra-hemispheric functional connectivity was discussed (right seed to right target; left seed to left target). Participant-level gPPI analyses were conducted separately for left and right auditory seeds using the AFNI program 3dDeconvolve. PPI regressors, the seed time series and the regressors of the original GLM model were included in the model.

## Resting-state functional connectivity

We also investigated intrinsic functional connectivity by analyzing resting-state functional data. Resting-state data were preprocessed using the GRETNA toolbox [62]. After removing the first 10 volumes, the remaining 740 volumes entered the following preprocessing steps: slice timing, realignment, spatial normalization, and detrending. Lastly, the normalized images were temporally filtered between 0.02 and 0.1 Hz, and covariates, including cerebrospinal fluid signal, white matter region signal, and head motion parameters, were regressed out.

Pearson correlation coefficients were calculated between voxel-wise time series in auditory dorsal stream regions, the same target ROIs used in gPPI analysis and the mean time series in bilateral auditory seeds to obtain the intrinsic functional connectivity in the auditory dorsal stream.

## ROI-based group analysis

To specifically investigate how aging and musical training modulate regional activities in the auditory dorsal stream during SIN perception, we performed the ROI analysis in BOLD activation, TiFC (beta coefficients for the PPI regressors), and RSFC. Voxel-wise TiFC, RSFC, and BOLD activation were extracted for each ROI. Then, voxels with the top 40% activation or connectivity levels across conditions were selected and averaged for the following statistical analysis [63]. Mixed-design ANOVAs (group*SNRs) were performed to investigate the effect of group and SNRs on neural activation or functional connectivity. FDR correction was applied for multiple comparisons across ROIs and post hoc analyses. Outliers, defined as data points beyond two standard deviations, were replaced with the groups' median. Correlation analysis was also performed to test brain-behavior correlation.

Standard parametric tests (e.g., ANOVA or linear mixed models) do not directly assess whether the magnitude of the difference between ONM and YNM exceeds that between OM and YNM. To determine whether the difference between YNMs and ONMs ($\Delta_{ONM-YNM}$) was larger/smaller than the difference between YNMs and OMs ($\Delta_{OM-YNM}$), we compared the two mean differences over left or right ROIs directly and employed a nonparametric bootstrap approach with 10,000 iterations to quantify the uncertainty in this comparison. In each bootstrap iteration, we resampled the data for each group with replacement, preserving the original sample size. For each resampled dataset, we recalculated $\Delta_{ONM-YNM}$ and $\Delta_{OM-YNM}$. From the resulting empirical bootstrap distribution, we derived the 95% confidence interval (CI) by taking the 2.5th and 97.5th percentiles. If this CI did not include zero, it indicated that the difference between YNMs and ONMs was statistically significantly larger/smaller than the difference between OMs and ONMs.

A robust linear mixed model analysis was further performed to investigate the group effect (e.g., ONMs versus YNMs) on neural indices regardless of brain regions, which showed significant group effects in each hemisphere. The formula was established as follows:

$$Neural\ index = group\ +\ (1|subject)\ +\ (1|brain\ regions)$$

The above analyses were performed using the packages "bruceR " and "robustlmm" in R [60,64].

### Inter-subject spatial correlation (ISPC) analysis

The TiFC of each voxel within the ROIs that showed significant group difference between OMs and ONMs under each condition was extracted for each participant. Then, we correlated the TiFC of each OM and ONM to that of each YNM for each ROI. The averaged correlation coefficients of each older adult to all YNMs represented the spatial alignment of the young. A higher spatial alignment measure denotes that the subject showed a higher TiFC spatial similarity as young adults. FDR correction was performed for multiple comparisons.

ISPC was also performed within the groups. That is, we correlated TiFC of each OM to that of other OMs for the ROI. The same procedure was also performed for the other two groups. The average correlation coefficients of each participant were calculated and compared to those of other participants within the group. Higher within-group ISPC denotes that the neural activities of the subjects in that group are more homogeneous with each other, which probably suggests that subjects in that group employ similar processing strategies.

### Activity spatial position analysis

Following ISPC, we performed the activity spatial position analysis within the specific ROI to examine whether older adults showed position differences of voxels that showed top TiFC. We extracted voxels that showed the top 10% TiFC strength of each participant within the ROI that showed significant ISPC group differences between OMs and ONMs for a more specific and targeted location analysis. Then, we averaged the x (left to right), y (posterior to anterior), and z (inferior to superior) axis coordinates of the voxels for each participant. T-tests were performed to compare the mean coordinates between groups. FDR correction was performed for multiple comparisons.

### Group analysis of voxel-wise BOLD and functional connectivity

We used ANOVA from AFNI program 3dMVM to examine group difference of voxel-wise BOLD activation and functional connectivity as measured with gPPI in auditory dorsal stream among the three groups (ONM, YNM, and OM). Model were constructed with the within-subject factor (SNRs) and between-subject group factors (ONM, YNM, and OM). Group activation and functional connectivity were tested using general linear model. Multiple comparisons were corrected using 3dClustSim ("fixed" version) with real smoothness of data estimated by 3dFWHMx (acf method) [65]. A total of 10,000 Monte Carlo simulations were performed to get the cluster threshold ($\alpha = 0.05$ family-wise error corrected). Results were visualized onto an inflated cortical surface using surface mapping with AFNI (SUMA).

## Supporting information

**S1 Text. Effect of years of education on the behavioral performance and neural responses in the audiovisual speech-in-noise perception task.**
(DOCX)

**S1 Fig. Illustration of fMRI trials and syllables.**
(TIF)

**S2 Fig. Voxel-wise functional connectivity as measured with gPPI in auditory dorsal stream seeded from left and right posterior STG for three groups.** $P_{fwe}$ < 0.05; STGpost, posterior superior temporal gyrus; OMs, older musicians; ONMs, older non-musicians; YNMs, young non-musicians.
(TIF)

**S3 Fig. Voxel-wise group comparison of functional connectivity.** Older non-musicians showed greater functional connectivity in left and right auditory dorsal stream regions than young non-musicians **(A, C)**, older musicians **(D)**. Older musicians showed greater functional connectivity in left auditory dorsal stream regions than young non-musicians **(B)**. $P_{fwe}$ < 0.05; STGpost, posterior superior temporal gyrus; OMs, older musicians; ONMs, older non-musicians; YNMs, young non-musicians.
(TIF)

**S4 Fig. Voxel-wise BOLD activation in auditory dorsal stream for three groups.** $P_{fwe}$ < 0.05; OMs, older musicians; ONMs, older non-musicians; YNMs, young non-musicians.
(TIF)

**S1 Table. Mixed-design ANOVA and post hoc analysis of task-induced functional connectivity.** The $F$ values and associated $P$ values represent the main effect of the group from a mixed-design ANOVA. The reported $P$ values have been corrected for multiple comparisons across the ROIs using the FDR method. The $t$ values and associated $P$ values are from post hoc pairwise comparisons with FDR correction.
(DOCX)

**S2 Table. One-way ANOVA and post hoc analysis of intrinsic functional connectivity.** The $F$ values and associated $P$ values represent the main effect of the group from a mixed-design ANOVA. The reported $p$ values have been corrected for multiple comparisons across the ROIs using the FDR method. The $t$ values and associated $P$ values are from post hoc pairwise comparisons with FDR correction.
(DOCX)

**S3 Table. Mixed-design ANOVA and post hoc analysis of BOLD activation.** The $F$ values and associated $P$ values represent the main effect of the group from a mixed-design ANOVA. The reported $P$ values have been corrected for multiple comparisons across the ROIs using the FDR method. The $t$ values and associated $P$ values are from post hoc pairwise comparisons with FDR correction.
(DOCX)

## Author contributions

**Conceptualization:** Lei Zhang, Yi Du, Claude Alain.

**Formal analysis:** Lei Zhang.

**Funding acquisition:** Yi Du.

**Investigation:** Lei Zhang.

**Methodology:** Lei Zhang, Claude Alain.

**Project administration:** Yi Du.

**Resources:** Yi Du.

**Software:** Lei Zhang.

**Supervision:** Yi Du, Claude Alain.

**Visualization:** Lei Zhang, Bernhard Ross, Claude Alain.

**Writing – original draft:** Lei Zhang.

**Writing – review & editing:** Lei Zhang, Bernhard Ross, Yi Du, Claude Alain.

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
