## [Editor Report · Decision Letter 0]

Dear Dr Zhang, 

Thank you for submitting your manuscript entitled "Cognitive reserves hold back age-related upregulation of neural activities in speech-in-noise perception" for consideration as a Research Article by PLOS Biology.

Your manuscript has now been evaluated by the PLOS Biology editorial staff as well as by an academic editor with relevant expertise and I am writing to let you know that we would like to send your submission out for external peer review.

Once your full submission is complete, your paper will undergo a series of checks in preparation for peer review. After your manuscript has passed the checks it will be sent out for review. To provide the metadata for your submission, please Login to Editorial Manager (https://www.editorialmanager.com/pbiology) within two working days, i.e. by Feb 07 2025 11:59PM.

Kind regards,

Christian

Christian Schnell, PhD

Senior Editor

PLOS Biology

cschnell@plos.org

---

## [Decision Letter · Decision Letter 1]

Dear Dr Zhang,

Thank you for your patience while your manuscript "Cognitive reserves hold back age-related upregulation of neural activities in speech-in-noise perception" was peer-reviewed at PLOS Biology. It has now been evaluated by the PLOS Biology editors, an Academic Editor with relevant expertise, and by several independent reviewers. 

In light of the reviews, which you will find at the end of this email, we would like to invite you to revise the work to thoroughly address the reviewers' reports.

As you will see below, the reviewers have mostly positive comments about your study, but ask for some methodological clarifications and justifications (including the choices for statistical analyses), better integration into the literature and more careful phrasing regarding the cognitive reserve. Reviewer 1 also suggests a few additional analyses and better presentation of the data. 

Given the extent of revision needed, we cannot make a decision about publication until we have seen the revised manuscript and your response to the reviewers' comments. Your revised manuscript is likely to be sent for further evaluation by all or a subset of the reviewers.

**IMPORTANT - SUBMITTING YOUR REVISION**

*Re-submission Checklist*

*Published Peer Review*

*PLOS Data Policy*

*Blot and Gel Data Policy*

Sincerely,

Christian

Christian Schnell, PhD

Senior Editor

PLOS Biology

cschnell@plos.org

REVIEWS:

Reviewer #1: This article by Zhang and colleagues reports an fMRI study that compares brain responses during syllable identification under noise conditions across three groups of participants: Older musicians (OMs), older non-musicians (ONMs) and young non-musicians (YNM). The research builds upon the broader context of investigating how cognitive resources derived from positive lifestyle factors, such as music learning, influence compensatory neural mechanisms associated with aging and observed in numerous previous studies. 

The authors compare two possibilities: the "Bolster Compensation Hypothesis" suggesting that the greater availability of cognitive resources increase compensatory activity in older adults during cognitive tasks; and the "Hold-Back Upregulation hypothesis" suggesting instead that high cognitive resources mitigate age-related decline in perception and cognition, thus decreasing compensatory activity compared to age-matched control participants. 

The authors consider a task of syllable identification (four choice) in noise, with three noise levels (-8db, 0 db, 8 db). As a measure of compensatory neural activity, the authors use the psychophysiological interaction between speech-related regions (posterior STG) and a set of parietal and frontal regions belonging to the dorsal auditory stream. Increased PPI is taken as a measure of increased compensatory activity, and thus an increase across groups (YNM < ONM < OM) is taken to support the first hypothesis, while intermediate levels for OMs (OM - YNM < ONM - YNM) are taken as evidence that older participants with greater cognitive resources are more similar than age-matched controls to young participants and thus support the second hypothesis. Reported behavioural and neural results support this latter hypothesis, highlighting the role of compensatory neural mechanisms in managing cognitive challenges associated with aging.

The study provides novel evidence suggesting that music training may play a significant role in improving neurocognitive resilience and it may be relevant for researchers studying the relationship between musicality and age-related cognitive changes. In general, the study is well-designed; however, several aspects require clarifications and improvements. These aspects concern the study description, which can be improved easily, but also relevant methodological choices, which now seem ad hoc and require further justification. I detail my considerations below: 

- Title (and scope): I would prefer a title that is closer to the experimental manipulation and experiments conducted, and thus I would replace "Cognitive reserves" with "Musical training" or "Musical expertise" as the equivalence Musical training = Higher cognitive reserves is an assumption that has not been tested in this study.

- The experiment, task and rationale for the analyses need to be introduced more extensively before the results description. Currently, it is not possible to follow the results section without first reading the methods section.

- Pg. 3, 77-81: Fig.1D should be Fig. 1C, Fig 1E should be Fig. 1D

- Pg. 3 91: SIN perception: "SIN" should be defined

- It is unclear if the behavioural data (and their analysis) refer to data collected during the fMRI measurements or outside the scanner.

- The SNR manipulation is interesting and the beahvioural results provide a perfect context for the analysis of the fMRI data. They show that OM performed worse than YNMs under all SNRs (as expected) but better than ONMs only under the intermediate and high SNR. I wonder why this SNR dependence has not been explored more extensively in the analysis of fMRI data as well. 

- The main analysis of the fMRI data relies on ROI-based PPI. However, PPI (and connectivity analyses in general) may be difficult to interpret without contextualizing the effects with a conventional voxel-wise analysis of brain activation and brain activation differences. 

- The authors only report regional activations in a suppl. Table. However, I would prefer to see the activation maps for the three groups and be able to judge whether these maps, at least t to inspect whether the activation across the three groups is qualitatively similar (that is are all ROIs active in the three groups?). 

Furthermore, I think that a GLM-based analysis of the voxel-wise between-group differences and the interaction between group and SNR could be very informative and should be presented to the readers.

-Related to the GLM analysis, is was unclear what are the "Six conditions of four syllables" (pg. 14, line 420). More in general, the experimental design, the timing of the stimuli and the separation of condition in blocks was not clear to me, and it could be improved (maybe with an illustration?).

- Pg 15, 450: Unclear: "Mixed-design ANOVAs (group*SNRs) were performed for A and visual enhancement"

- For the PPI analysis, it is stated "PPI regressors, the seed time series, and the regressors of the original GLM model were included in the model". What did enter the ANOVA analysis? Are the betas for the PPI regressors? Were they significant? Or, in other words, did the PPI regressor explain additional variance that was not explain by the original GLM predictors?

- Why an ad hoc non parametric analysis is needed to test the contrast (ONM - YNM <> OM - YNM)? Can't this be done in the context of the prametric statistical framework of the main analyses (ANOVA, linear mixed effects modeling)?

- The analysis based on the spatial alignment of the connectivity patterns needs a rationale justification. I am not convinced that some shift in the mean coordinate of a few voxels across groups can be easily interpreted? What would be the neural substrate for this observation?

Reviewer #2: Dear Authors,

thank you for the wonderful work done! Nevertheless, some additional questions and comments:

1) Introduction should include more detailed explanation why you expect differences between groups based on occupation and age. It is unclear in the current text. 

2) I would argue that, while occupation (in this case - musician) is an element of cognitive reserve, it can not be used as the only measure of CR. Even more, within the introduction, authors introduce different models of cognitive compensatory ageing, but omit The Cogntive Reserve theory, providing only references to aged publications by Stern et al. If authors are adamant in keeping occupation-musician as the only measure of cognitive reserve, I would definately suggest integrating more information on the actual theory. In addition, current consensus on cognitive reserve is that it should be (a) measured longitudinally and (b) considered as a moderator between the brain and cognitive functioning in case of pathology. Please, refer to the publication by Reserve and Resilience Framework Group: Stern, Y., Albert, M., Barnes, C. A., Cabeza, R., Pascual-Leone, A., & Rapp, P. R. (2023). A framework for concepts of reserve and resilience in aging. Neurobiology of aging, 124, 100-103.

3) Comments and recommendations regarding participants: 

- did you also consider specific areas of music? There are studies suggesting instrument-specific activities.

- considering the size of the sample, replacing outliers with the group average might not be the best approach, as it increases the risk of Type I error (e.g. see Gress, T. W., Denvir, J., & Shapiro, J. I. (2018). Effect of removing outliers on statistical inference: implications to interpretation of experimental data in medical research. Marshall journal of medicine, 4(2), 9. and Vankov, I. I. (2023). The hazards of dealing with response time outliers. Frontiers in Psychology, 14, 1220281. for differences in outlier treatment methods). While I do not claim this is the case, my suggestion would be to triangulate the data analysis using different approaches to removing outliers, to ensure that the correct approach has been chosen.

4) Regarding data analysis:

- while exclusion of education as is discussed in the supplementary text, it should be discussed more in the body of the text as well. It should also be clarifies, what is meant by "the self-reported years of education cannot represent the education level of the older subjects reliably". It might not represent the level of knowledge, however, it can quantify the formal and informal education, which is the most common way of measuring education (even if not the best). In any case, I would suggest to briefly clarify this in the body of the text as well.

- In general, Results section is well structured and appropriate data analysis methods are used.

5) Discussion: 

- while the authors have presented their results within the context of the current literature, the limitations of the study are not really discussed. I would encourage the authors to address the impact of the issues like sample size and characteristics, research design, task type etc. on the study results.

- to add to the comment at the beginning, the concept of cognitive reserve in the discussion section is used very loosely. 

Overall, this is a well-structured article with well conducted data analysis, however, some work on the theoretical framework is suggested.

---

## [Decision Letter · Decision Letter 2]

Dear Dr Zhang,

Thank you for your patience while we considered your revised manuscript "Musical expertise holds back age-related upregulation of neural activities in speech-in-noise perception" for publication as a Research Article at PLOS Biology. This revised version of your manuscript has been evaluated by the PLOS Biology editors, the Academic Editor and the original reviewers.

Based on the reviews and on our Academic Editor's assessment of your revision, we are likely to accept this manuscript for publication, provided you satisfactorily address the following data and other policy-related requests:

* We would like to suggest a different title to improve its accessibility for our broad audience: 

Long-term musical training can protect against age-related upregulation of neural activity in speech-in-noise perception

* Please add the links to the funding agencies in the Financial Disclosure statement in the manuscript details.

* Please include the approval/licences number of the ethical approval of this study.

* Please include information in the Methods section whether the study has been conducted according to the principles expressed in the Declaration of Helsinki.

* Please move supplementary methods, discussion and references to the main manuscript file. We do not have a word count limit for methods.

* DATA POLICY:

Regardless of the method selected, please ensure that you provide the individual numerical values that underlie the summary data displayed in the following figure panels as they are essential for readers to assess your analysis and to reproduce it: 2, 3C, 4BEF and 5.

* CODE POLICY

We expect to receive your revised manuscript within two weeks. 

*Published Peer Review History*

*Press*

Sincerely,

Christian

Christian Schnell, PhD

Senior Editor

cschnell@plos.org

PLOS Biology

Reviewer remarks:

Reviewer #1: I thank the authors for their response and for the changes implemented in the manuscript. The manuscript can be accepted for publication.

---

## [Editor Report · Decision Letter 3]

Dear Dr Zhang,

Thank you for the submission of your revised Research Article "Long-term musical training can protect against age-related upregulation of neural activity in speech-in-noise perception" for publication in PLOS Biology. On behalf of my colleagues and the Academic Editor, Laura Lewis, I am pleased to say that we can in principle accept your manuscript for publication, provided you address any remaining formatting and reporting issues. These will be detailed in an email you should receive within 2-3 business days from our colleagues in the journal operations team; no action is required from you until then. Please note that we will not be able to formally accept your manuscript and schedule it for publication until you have completed any requested changes.

PRESS

We frequently collaborate with press offices. If your institution or institutions have a press office, please notify them about your upcoming paper at this point, to enable them to help maximize its impact. If the press office is planning to promote your findings, we would be grateful if they could coordinate with biologypress@plos.org. If you have previously opted in to the early version process, we ask that you notify us immediately of any press plans so that we may opt out on your behalf.

Sincerely, 

Christian

Christian Schnell, PhD

Senior Editor

PLOS Biology

cschnell@plos.org